# Snakebites in the Central American Region: More Government Attention Required

**DOI:** 10.3390/tropicalmed10080225

**Published:** 2025-08-12

**Authors:** Eduardo Alberto Fernandez, Ivan Santiago Fernandez Funez

**Affiliations:** 1Department of Health Sciences, Faculty of Applied Health Sciences, Brock University, St. Catharines, ON L2S 3A1, Canada; 2Faculty of Applied Health Sciences, Brock University, St. Catharines, ON L2S 3A1, Canada

**Keywords:** snakebites, Central America, *Elapidae*, *Viperidae*, envenoming, clinical manifestations, demographics, healthcare access

## Abstract

A review was conducted on snakebites in Central America. Information was extracted using the databases of PubMed, SciELO, and LILACS. Information included retrospective studies, case reports, and case series; in this way, valuable information was retrieved from limited sources. The identified studies comprised those discussing envenoming snakebites. Several species were identified, but three of them had major epidemiological features impacting envenoming by snakebites: *Bothrops asper*, *Crotalus simus*, and *Micrurus* sp. Adolescents and young adult males living in rural areas and engaged in agricultural activities were identified as the main victims of snakebites by clinical records. Symptoms of local damage in the bite sites included edema and skin and muscle necrosis. In addition, the cardiovascular system was affected, with symptoms like hypotension, bleeding, and coagulation disorders. Neurotoxicity causing sensitivity and motricity problems was also reported. For El Salvador, accidents caused by *Crotalus simus* and *Micrurus* spp. were given more attention due to their greater relevance. The role of *Bothrops* species was more relevant in the envenoming reported by other countries. Treatment was found to be provided based on antivenoms produced in Costa Rica, and the recovery of the patients depended on the time elapsed between the accident and the initial treatment in the healthcare system.

## 1. Introduction

Snakebites were included in 2017 as part of the neglected tropical diseases to be addressed as a priority by the Sustainable Development Goals. Snakebites need to be incorporated as a priority in the public health initiative because of their emerging importance. Snakebites occur because of the overlapping of humans and snake habitats caused by activities such as agriculture, pasturing, and livestock farming in different parts of the world. Commercial farming has opened vast areas of land in tropical and subtropical regions for human activity, and unplanned encounters of humans and snakes (venomous and non-venomous) occur because of this. Consequentially accidental bites can occur, causing disease, death, and disability [1,2].

According to recent WHO figures, there are around 5.4 million cases of snakebites per year worldwide, and from those, close to 2.7 million develop clinical manifestations of envenoming [3,4]. The greatest proportion of venomous snakebites is reported in South Asia and sub-Saharan Africa and represents an important part of their healthcare demand. The Americas and Australia also report snakebites, and their response to this health problem varies broadly [2,3,4], as reflected by their surveillance program and the management of cases and mortality.

Global estimates include information for Central American countries (see Table 1) regarding mortality caused by snakebites, death age standardized rates (DASR) for this cause, and the percentage change in a period of 29 years. An additional indicator is years of life lost (YLLs), which is the measure used to quantify the impact of premature mortality on a population [5,6].

According to Harrison (2009), people dedicated to farming activities and extraction of silviculture products have an increased risk of coming into contact with poisonous snakes, as their hands, feet, and legs are in contact with areas where snakes live or move in forested areas [8].

The next section will present some general aspects of snakebites in Central American countries and their populations.

### 1.1. Snakebites in the Central American Region

Central America is an isthmus between North America and South America, with access to the Caribbean and the Pacific coasts; it includes Belize, Costa Rica, El Salvador, Guatemala, Honduras, Nicaragua, and Panama. In Central America, variations in climate conditions lead to the presence of different venomous snake species, causing different clinical manifestations (depending on the different components of their venoms) [9].

According to Gutiérrez (2016) [10], a large proportion of venomous snakebites are caused by *Bothrops asper* in Central America. Envenoming caused by this species has been better documented in Costa Rica; information from other Central American countries is available but less complete [9,10].

*Crotalus simus* has been reported as one of the main venomous snakes in inland and dry forest areas, while coral snakes of the genus *Micrurus* (family Elapidae) are more relevant as a cause of snakebites along the Pacific coast, being associated with 1% of the total snakebites [11,12].

### 1.2. Occupational Risk for Snakebites (Ophidian Accidents) in Central America

A large proportion of venomous snakebites in Central America are associated with agricultural activities and can be considered under the scope of occupational risk, with the affected population having an increased vulnerability to this condition and its complications [13].

Morbidity and mortality caused by snakebites in Central America is attributed to snake species grouped into two families: Viperidae and Elapidae. Cases are not fully reported because of limited accessibility to health services and weakness of the epidemiological systems. The specifics on the type of snake populations associated with every case are lost because of deficient surveillance reporting and personal reactions associated with the encounter with these reptiles [14,15]. A previous study from one of the authors mentioned the difficulties in surveillance and how local publications are sometimes the only source of information. We have expanded our search to those additional publications [14].

### 1.3. Species of the Family Viperidae

*Bothrops asper:* This is the main representative of this family in Central America. The envenoming caused by *B. asper* can cause damage to several organs and systems, but those described as more relevant are at a local level. Cytotoxic effects cause extensive tissue destruction in the area around the primary bite site, compromising cutaneous, subcutaneous, and muscular tissue [13,14].

Muscle damage is a frequent outcome resulting from local regeneration deficiencies, muscle loss, and fibrosis. This damage results in varied levels of disability [15]. The deficient regeneration of the muscle remains a matter for research; one of the hypotheses is that snake venom metalloproteinase (SVMP)-induced basement membrane damage in micro-vessels, muscle fibers, and nerves is the main culprit for the poor regenerative outcome [15,16]. The cardiovascular system is the most affected.

*Crotalus simus*: Venoms from adult specimens of *C. simus* from Costa Rica present high proteolytic, hemorrhagic, and edematogenic activity and are devoid of neurotoxic activity. Local symptoms may be severe, with pain, massive swelling, blistering, and necrosis [16,17].

Systemic effects are relatively mild or moderate, involving hemostatic disturbances (hypofibrinogenemia), spontaneous systemic bleeding, and renal failure. Neurotoxicity may also occur, but it is less frequent [16,17,18,19].

### 1.4. Species of the Family Elapidae

The envenoming by local Elapidae species like those in the genus *Micrurus* presents neurotoxic manifestations provoked by neuromuscular blockage predominantly induced by post-synaptic-acting low-molecular-mass neurotoxins, as mentioned by Gutiérrez [20]. Some of these clinical features include initially mild pain, nausea, vomiting, and dizziness. As the venom spreads, more severe symptoms can appear, such as muscle weakness, paralysis (descending paralysis with bulbar findings appearing first), difficulty breathing, and neurological issues like slurred speech and drowsiness [21,22].

Some of the most relevant snake species in Central America are presented in Figure 1.

### 1.5. Treatment of Snakebites

The health systems in Central America use antivenoms produced in laboratories using a mixture of the venoms of the most common venomous snakes in the region. The process of antivenom production requires the inoculation of increasing doses of venom in laboratory animals to produce antiserums. The antiserums are then processed and purified for use in individuals suffering snakebites to neutralize the venom [23].

The laboratory in charge of the production in Central America is in Costa Rica, which uses venoms of local snake species (to make them more specific to the chemical structure, producing a more complete neutralization of the toxin effects) [24] (See Table 2).

Some antivenoms are imported from Mexico, Argentina, and other laboratories in South America, which can be effective in attenuating the toxin effects as well. Each country assigns antivenom supply to hospitals of secondary and tertiary levels (located in main cities), where the product can be kept under low temperature (cold chain). However, these facilities are sometimes located a few hours away from the critical spots (in rural and distant areas) where most snakebites occur. This represents an obstacle for its opportune use and reduces its impact on the toxin effects. As a shared problem, due to its cost and cold-chain requirements, many countries do not have enough antivenom for their at-risk population [25].

The most distant communities use natural products (derived from plants and animal structures) known to have an effect on some symptoms, such as pain and inflammatory reactions, but with less tested effect on other clinical manifestations, such as alterations on coagulation, hemorrhages, effects on distant organs, and neurological symptoms (peripheral and central) in the case of *elapid* toxins (in coral snakes). There have been recent efforts to scientifically study the effect of some active ingredients of plants on venom. In Central America, there has been a particular interest in identifying the effect of ethnomedicine products on the hemorrhagic effects of *Bothrops asper* given their role in envenoming; however, the results have shown very limited success [26].

The authors of this study want to build on the findings of a previous Americas-wide study, where we found important gaps in epidemiological surveillance and how information relies on studies conducted by academic or international organizations. A pattern of distribution of snakebites was found [14].

The current study is designed to identify the scientific knowledge on the topic of snakebites and determine how each country has found the demographic characteristics of the affected population, the ecological characteristics of areas affected by different venomous snakes, and the accessibility problems of reaching opportune medical attention. It is important to know about the findings gathered by different studies conducted in this region and their coincidences and differences in order to increase knowledge of the problems and needs, thereby helping develop appropriate government and social responses.

## 2. Methods

The issues to be addressed included the main epidemiological features of snakebites in the countries of Central America and the time elapsed between the snakebite and initial healthcare access.

A narrative review of recent publications on the epidemiology and clinical characteristics of snakebites published in Central America was conducted to identify common species of venomous snakes, the symptomatology reported in the cases, and the demographics of the population affected by the event.

Different databases were reviewed, including PubMed, SciELO, and LILACS. Studies published in English and Spanish were included. Given the limited number of publications in the region, we included publications from 1950 to the current year as well as publications retrieved from university journals in the region where they were available.

Retrospective studies, case reports, and case series from countries in Central America were included, and the objective was to identify the common symptoms and correspondence to the snake species and their main toxins. Further, the casuistic data gathered by researchers in different countries, including the demographics, symptomatology, and outcomes, are described.

## 3. Results

Different authors provided very similar information on the demographics of the victims of snakebites (similar patterns). A total of 32 papers were reviewed in this comprehensive narrative review.

There was no scientific confirmation of the species causing the envenoming, with only self-reports of the common names of snakes available; frequently, even that information was absent. The symptoms were described in a similar way across different countries’ studies, and some identified patterns of ophidian accidents and responsible species according to ecological and climactic areas.

### 3.1. Belize

We did not find specific publications referring to snakebites in Belize, and the government does not issue information on this topic. However, there are educational materials for military personnel about the type of venomous snakes, including *Bothrops* species and different Crotalids and Elapidae snakes.

In the local press, they mentioned an attack by an unidentified snake on a 14-year-old boy, with a bite on the leg causing local effects. He was treated in a nearby clinic without any further complications beyond local symptoms like edema, reddening, and local lesions on the patient’s leg [27]. Estimates were calculated by GBD and are included in Table 1.

### 3.2. Guatemala

Six different publications were identified for Guatemala. Wellmann (2020) [28] conducted a review of different studies carried out in Guatemala, finding that venomous snakebites were reported as causing a series of hemorrhagic manifestations and coagulopathies. Additionally, manifestations of compartment syndrome, renal failure, and shock were reported in the most severe cases. The necropsy reports of patients who died (a subsample of seven) showed clinical findings of cerebral hemorrhages, renal necrosis, and hepatic cirrhosis. The study authors had difficulties trying to identify the responsible species for many snakebites, but those identified included basically four species: *B. asper*, *Agkistrodon bilineatus*, *Porthidium ophiomegas*, and *C. durissus* [28,29,30].

Letona (2012) [31] mentioned *B. asper*, *C. simus*, *M. nigrocinctus,* and *Atropoides mexicanus (A. mexicanus)* snakes as responsible for ophidian accidents in Guatemala. Medical reports included 7377 snakebites from 2001 to 2010, most of them from locations in the lowlands of Guatemala. *Bothrops asper* was the main species responsible for most snakebites in northern Guatemala (in the Caribbean basin), while *Crotalus simus* was more frequent in the southern part (Pacific basin). Envenoming cases caused by snakes of the genus *Micrurus* were few, with two reported in 2017 [28].

The victims of snakebites described by Letona were more frequently in the age range of 10–19 years, with most of them being males living in rural areas and dedicated to farming activities, but some suffered the bite at home or close to it. Women suffered a lower number of bites doing domestic work, collecting lumber, and getting water for their family needs [30].

A different study conducted by Guerra-Centeno (2016) [32] analyzed 305 clinical charts of victims of snakebites. The charts corresponded to two regional hospitals of Guatemala during the years 2008–2013, with one of them serving the population of the central and southern provinces of Guatemala (Hospital Escuintla) and the other hospital (San Benito Peten) serving patients of the northern and central parts of Guatemala.

A total of 187 patients were men (61.3% of the total) and dedicated to agricultural activities; 32% of the victims were farming while they experienced the event, 27.8% were at home, and a smaller proportion suffered the event while walking on the road or paths between home and agricultural fields. Snakebites were reported as occurring with similar frequency during daytime and nighttime. The mean age of the victims was 25.2 years. The lower limbs were the body parts most frequently bitten [31].

The mean time to reach the hospital after the event was 5.6 h, and the author found it was related to difficulty in transportation, remote location of incident occurrence, economic factors (no money to start mobilizing to the hospital), and the stops taken to look for assistance by traditional healers and natural medicine people [31].

The study identified that both hospitals had differences in the main snake species reported as responsible for the event (snakebite). The northern hospital (San Benito Peten) identified 88.5% of the clinical cases related to *Bothrops asper* snakebites. In the southern hospital, *Bothrops asper* was reported with less frequency (13.3%), while *Crotalus simus* was responsible for 28.8% of cases, *Agkistrodon bilineatus* was associated with 37.5% of the events, and coral snakes (*Micrurus* sp.) were reported in 13.3% of cases [31].

From the findings in the reported studies, we consider that the epidemiology of the snakebites in Guatemala seems to correspond to descriptions given by authors in other countries in the Central American region. Similar human and institutional characteristics can be an explanation for these similarities [22,32].

### 3.3. Honduras

In Honduras, there were reports of snakebites caused by the most common species, and there were a few case reports reflecting the natural history of snakebites after 24 h. The studies were conducted through the review of clinical charts in six hospitals, with four representing local and regional coverage and two representing national coverage.

A total of fourteen studies were identified, three of them in the pediatric population and the rest in the general population. Those with a pediatric population had a median age of 11 years, and those in the general population had a mean of 18 years. A common pattern in the studies was the hemorrhagic symptoms (in seven hospitals), followed by pain and edema, and neurological manifestations in one patient. There were more male than female patients exposed to snakebites [33,34,35,36,37,38,39,40].

The hospitals in the northern regions had more reports of snakebites caused by *Bothrops asper*, while in the rest of the country, cases were distributed among other *Bothrops* species, *Crotalus simus*, and other *Crotaline* species. Less than 5% of cases had a record of encounters with coral snakes *(Micrurus* sp.) [33,34].

As described in studies conducted in other Central American countries, the victims of snakebites came from rural locations and were dedicated to farming or domestic activities or were students walking on rural roads and paths [33]. Some of the barriers to opportune attention in the hospital were the distance to the health clinic and factors related to accessibility of healthcare. These factors affected the critical time to be treated for envenoming because the venom action depends on the time the toxin is in circulation and in the local tissues. Two studies marked a range of 3–5 h, and two more provided a mean of 5.4 h to reach the healthcare place [22,36].

A case reported an envenoming event after a *Bothrops asper* bite, and the most relevant manifestations included headache, dizziness, weakness, epistaxis and paresthesia, distal coldness, prolonged 5-s capillary filling, and dry mucous membranes, with the patient’s condition evolving after the second day to tachycardia, tachypnea, dyspnea, pleural effusion, and hemothorax. The patient had improvement in his condition after treatment, including the use of antivenom [37].

Patients described in the different studies tended to be young, male, and rural dwellers, usually farmers who suffered bites in the limbs, especially the lower ones, with symptomatology of haemato-toxicity [38,39]

In Santa Teresa, a hospital located in the central dry valley of Comayagua, reports of snakebites identified the causal snakes as *Crotalus simus*, *Micrurus* sp., *Bothrops nasutus,* and *Bothriechis marchi,* with a smaller number of snakebites by *Bothrops asper* [35].

An author of agricultural topics found that in the western part of Honduras, *Crotalus simus* and *Bothrops asper* are the most common causes of snakebites. He emphasized that envenoming is a concern in human and veterinary health. The study was conducted in the community in extra-hospital settings, and the community informants mentioned three deaths during 2011 among farmers having livestock on their farms [40].

### 3.4. El Salvador

El Salvador is a country located in the dry Pacific corridor of Central America. *Bothrops asper* is not found here, with the main cause of snakebite envenoming attributed to *Crotalus simus*, while there are other snakes with a secondary role in these events. Importantly, El Salvador is in a part of the Pacific basin with no humid forest [41].

According to statistics kept by the Ministry of Health in El Salvador, 1130 snakebites were reported by the National System of Information on Morbidity and Mortality for the period 2014–2019. The mean was 188.3 cases per year, and the range was 161 to 215 cases. Snakebite is an obligatory notifiable event, and snakebites are reported regularly [42].

A different study used data from the Epidemiological Surveillance System and was conducted by Chirino Molina (2025) [43], who used information from 2011 to 2022. She included “all people of any age and sex bitten by a poisonous snake”. The cases included in the study presented a clinical condition of progressive edema around the area of the bite, dizziness, and hypotension (mild to severe) with or without any of the following symptoms: hemorrhages, paresthesia, necrosis in the bitten area, and ptosis (mono- or bi-palpebral).

The study included 1503 records, and after excluding duplicates and files of foreign patients, the final total was 1472 records. Demographic information was included in a new database (Microsoft Excel 2019) and a key variable—time between the event (snakebite) and clinical attention—was included [42].

A total of 61% of the patients were male, and 83.2% came from rural locations. The patients’ ages were in the range of 1–98 years, and the median was 28 years for men and 27 years for women. Patients were older than in other Central American countries [42].

Greater frequency of reported snakebites occurred from May to September, which corresponds to the rainiest season in El Salvador.

The Chirino Molina [43] study identified *Crotalus simus* and other species such as *Porthidium ophriomegas*, *Cerrophidion wilsoni,* and *Micrurus nigrocinctus* as being responsible for snakebites, specifically in clinical records [43].

Gutiérrez (2020) [41] described that in the period 2014–2019, the official records included four deaths and a case fatality rate of 0.44%. The incidence of snakebites and mortality because of this cause were considered the lowest in the Central American region [41,43].

### 3.5. Costa Rica

Costa Rica has identified *Bothrops asper* as the main cause of snakebites in the country, affecting young adults and causing local effects and hemorrhagic manifestations, edemas, hypotension, and other systemic disorders (cardiovascular shock and acute renal failure).

*B. asper* has been associated with wet lowland regions in Costa Rica as well as in countries with similar ecological characteristics. Previous studies have demonstrated a higher incidence of this neglected disease in the last years of the twentieth century and the first years of the current century. From 1993 to 2006, there were 48 fatalities due to snakebites. Mortality rates ranged from 0.02 per 100,000 population in 2006 to 0.19 per 100,000 population in 1993. The most affected age groups were 20–29, 40–49, and 50–59 years, and fatal cases were predominantly seen in males over females by a ratio of 5:1 [44].

A recent study completed by Sasa and Segura (2020) [45] in six hospitals reviewed the charts of a total of 475 victims of snakebites who attended the selected hospitals of the country in 2012 and 2013. The incidence rate for the country during the two studied years was 9.44 and 10.76 per 100,000 inhabitants per year, and the incidence of snakebite differed between months (χ^2^ = 30.93, df = 11, *p* < 0.001), showing a peak in May–July and another in October–November. This study showed an association with agricultural activities and a much lesser association with recreational activities, but accidents could occur in peridomiciliary spaces as well [45].

Historically, Costa Rica has prioritized better snakebite management. At the same time, there has been an increase in antivenom production and public awareness of this ophidian accident in the regions of the country [46].

### 3.6. Panama

In Panama, Jutzy et al. (1953) reported a series of 23 patients who suffered snakebites caused by *Bothrops* spp., and in seven cases, the outcome was fatal, with shock and hemorrhages of the central nervous system [47].

Panama was considered the country with the highest incidence of snakebites in the Central American region by Vélez (2017) [48]. The authors of the paper identified characteristics of the venoms in four regions of Panama, with a profile of lethal, hemorrhagic, in vitro coagulant, defibrinogenating, edema-forming, myotoxic, and indirect hemolytic activities, with subtle quantitative variations between samples of some regions [48,49].

Human accidents caused by *B. asper* are characterized by prominent local tissue damage, i.e., edema, myonecrosis, dermonecrosis, blistering, and hemorrhage, and systemic manifestations, i.e., coagulopathies, bleeding in various organs, hemodynamic alterations that can lead to cardiovascular shock, and acute kidney injury [49,50].

A study conducted in the Veraguas province found that *Bothrops asper* is responsible for nearly half of the reported snakebites, and the rainy season is the time with higher frequency, with the venom causing frequent damage to feet, toes, and hands, in accordance with the areas in closer contact to the bite [51].

Another study analyzed the effects of the reference venom of *B. asper* from Panama at local and systemic levels, and the results were consistent with the venoms of the Viperidae family. This venom possesses lethal, hemorrhagic, myotoxic, edema-forming, defibrinating, and in vitro coagulant activities. This toxicological profile is like the one previously described for *B. asper* venoms from other Central American countries as well as from Mexico [52].

### 3.7. Nicaragua

Several studies have been conducted in Nicaragua, including one by Campbell and Lamar, who found the main species involved in snakebites were *Crotalus simus and Bothrops asper*, and most fatal events were reported in the east and center regions [53]. Similar findings were stated by Hansson (2010) [54], who also found a 5-year incidence of 56 snakebites per 100,000 inhabitants, 34 reported fatal snakebites in Nicaragua in 2005–2009, 0.6 fatal cases per 100,000 inhabitants in 5 years, and a 1% case fatality rate [51].

Seasonal variations, which are most pronounced in the eastern part of the country, where the incidence almost triples between the lowest (May) and highest (December) months, were described by Hansson in the 2010 publication [54].

Moreno Avellano (2000) [55] described people aged between 12 and 20 years old, living in rural areas, and working in agriculture as the most affected groups, with a clear male predominance. The lower limbs were affected in 71% of the cases, and the upper limbs in the rest; the mean time to reach a hospital was 8 h, and the mean stay in the hospital was 4 days in a recovered condition, with 89% of them receiving treatment with antivenom. The most common snake species in his study was *Crotalus durissus*. The most common symptoms included pain, edema, bleeding, paresthesia, and vomiting [55].

Studies in Central America have been oriented towards clarifying the epidemiology of snakebites, but some of them have provided information on mortality without placing emphasis on some epidemiological parameters. We selected some of them to prepare the following table (see Table 3). It is expanded to the 42 studies in Appendix A.

Data were taken from the selected publications cited in this paper.

We added a map as illustration for the findings in every country of Central America (Figure 2). 

## 4. Discussion

Snakebites constitute an important health problem in the Central American region, but they still need to be studied and described in a more complete way.

As a neglected tropical disease, snakebites suffer from a lack of interest by most governments in Central America. The cases occur in distant towns far from decision centers, and the pressing demand for solutions is low, which contributes to the postponement of effective prevention and control policies [13].

While other world regions are prioritizing the problem, Central America is traditionally focused on communicable diseases that affect large urban centers with industrial or commercial importance or vector-borne diseases that can paralyze economic activity [6].

The problem has been perceived as a concern for agricultural activity and perceived only as a work accident that can be addressed locally by individuals, their families, and employers (if they take part in high-scale agricultural activity), even though the impact can be long-term [56].

The recent interest has been stimulated by the international movement to reach the Sustainable Development Goal of reducing the burden of neglected tropical diseases, which include snakebites [56]. However, tackling the problem of snakebites requires specifying the agenda to reduce the problem and identifying stakeholders that can enhance effective action to reduce morbidity. This includes empowering communities, as is happening in Brazil, Africa, and India [1,57,58].

Most countries in the area need to develop plans to improve accessibility to primary medical care and antivenom treatment. According to the different reviewed publications, the time elapsed between the snakebite and the arrival at health facilities can take from several hours to longer than one day, enough time for the venom to destroy tissues around the bite zone and cause hemorrhages or coagulation disorders. The damage may affect multiple organs, even in the central nervous system, by causing endocranial hemorrhages and even sensitive and sensorial damages in the case of snakes of the Micrurus genus, leaving individuals disabled and resulting in low quality of life for them and their families.

From the history of these countries, the relationship between subsistence agriculture and the emergence of commercial agriculture, especially in the Caribbean Coast, where *Bothrops asper* lives, is clear. Agricultural activity brings human and snake habitats closer, leading to undesirable encounters and accidents [12,48].

It is important to have a population that is better informed about the types of endemic snakes and the expected clinical damage of such ophidian accidents. Knowledge of the risk posed by snakebites seems to be unevenly distributed in the population of different countries. Costa Rica should be mentioned as the country with more advantages because of the local production of antivenoms, access to clinics, and better knowledge among the population [25].

Two areas that need to be considered as priorities are an adequate supply of antivenom where they are required (according to the epidemiological data); the frequency of accidents caused by different species to avoid problems; and the discarding of unused antivenom due to lack of use, inadequate storage, or them not being the kind required in the local clinic.

Knowing that access to medical facilities is difficult in many locations, many people rely on local treatments, usually based on plant extracts. These products require more research of their real benefit. In cases where there is a body of knowledge supporting it, processes of knowledge translation should be conducted.

Information taken from patients needs to be accurate and complete (as much as possible) to develop an adequate profile of the victims of snakebites (ophidian accidents, as known in this region) to feed the decision-making of the health systems in the countries affected by this neglected disease.

An important element is to recover information of mortality that may be hidden from the health system because the victims do not reach the hospital or the death(s) do not cause a sanitary action. Some countries have recovered a small proportion of these mortalities, as in the case of Panama, Costa Rica, and Nicaragua.

Not all patients receive treatment with antivenom, and according to optimistic calculations, the coverage reaches less than 80%. One of the reasons for this deficit in treatment is the lack of clarity of the protocols for management of these patients or the total absence of them in medical facilities [25].

Since this health condition is mediated by encounters with venomous snakes, it is important to inform adults and educate children about the risks of encounters with these reptiles; familiarize the population with the features of the most common venomous snakes in different regions of Central America; and, if possible, desensitize the population to the initial emotional impact. Only then will the victims have the ability to identify the snake type. This fact can facilitate the early and specific management of cases and prevent these accidents.

Clinical records usually keep information on snakebites and their immediate management in medical facilities, but the sequelae to the envenoming are not well recorded. Conditions like physical disability and mental trauma can remain as an individual and social problem, with the impact neither qualified nor quantified, although there are some estimates by international organizations.

There are acute consequences of snakebites in productivity and school attendance. Sometimes, when treatment does not solve all the physical problems, there is a reduction in the capacity to perform daily functions in daily life and participate in economically productive activities. In Central America, and especially in the most distant communities, mortality and disability are difficult to avoid and treat [53]. The ministries of health in Central America must address this problem of neglected population, as has been achieved in other regions of the world [3,54].

## 5. Conclusions

Snakebites constitute a worldwide problem, and Central America is not an exception to this neglected problem, but it is not well known there except in small research groups. The increasing overlap of the habitat of multiple snakebite species and the rural habitat of the human population has led to an increased risk of suffering snakebites. Health services need to respond to ophidian accidents by understanding the epidemiology of the problems; analyzing interactions between the reptiles and communities that follow a similar pattern across Central American countries; observing the coincidence of different ecosystems and the relative predominance of certain species and patterns of clinical symptoms; and realizing the need to be ready with more specific antivenoms to prevent tissular damage, bleeding disorders, and neurological impact, which are among the most frequent clinical manifestations.

With the exception of El Salvador, the most frequent ophidian accidents are caused by the attack of the *Bothrops asper* snake (its preferred habitat is the humid lowlands along the Caribbean coast and some inland areas with similar characteristics).

From the data reviewed by researchers in different countries, the anatomic areas with the most frequent exposure to snakebites are the limbs, especially the lower limbs (feet, legs, and occasionally thighs). The upper limbs are exposed through the manipulation of tree branches, bushes, soil, or crops during agricultural routines. Any preventive program has to devise methods to protect or cover those areas, starting with the avoidance of vulnerable sandals or being barefoot in areas of risk.

Given the remoteness of many villages and hamlets affected by snakebites, Central America needs to bring treatment facilities closer to these rural areas and learn more about the traditional practices used for treating snakebites to verify their real value and the need to systematize them to provide effective care to these less accessible localities.

### Future Directions

The governments in Central America need to prioritize the problem of snakebites as part of the third Sustainable Development Goal related to neglected tropical diseases, not only for the health sector but also for education and productivity, as they are also affected by this type of accident. Management of snakebites as an occupational risk and disease needs to focus on preventive measures prior to any exposure and preparation for what to do once accidents have occurred.

The academic sector needs to stimulate more research on the ecology of the disease and the toxinology of the main reported venoms in the region. From a clinical perspective, more research is needed on disease reporting in the population and active promotion of the collection and analysis of venoms to produce more specific antivenoms.

Accessibility to antivenoms for the population requires identifying areas closer to the most frequent places of accidents and training medical personnel in those sites to use antivenoms according to official guidelines.

The study of natural products is already encouraged in some countries, but it is necessary to stimulate this type of study in graduate programs to identify alternatives.

The aspiration for governments and communities should be to have a world free of mortality caused by ophidian accidents and to develop preventive actions to avoid morbidity.

Referenced websites in the figures and Table 2:http://gisgeography.com/central-america-blank-map/, accessed on 20 May 2025http://www.inaturalist.org, accessed on 20 May 2025https//icp.ucr.ac.cr/en/services and products/products-human-use, accessed on 20 May 2025https://birmex.gob.mx/, accessed on 20 May 2025https//www.miamidade.gov/fire/library/antivenom-species-covered.pdf, accessed on 20 May 2025

## Figures and Tables

**Figure 1 tropicalmed-10-00225-f001:**
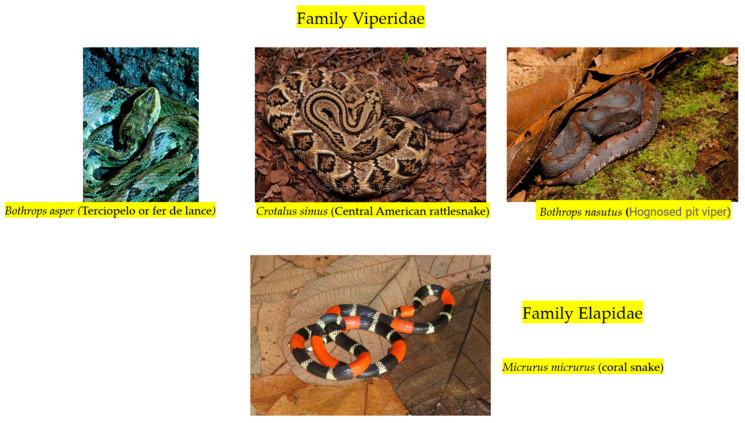
Common venomous snakes in Central America described in the paper. Images taken from the Naturalist website (https://www.inaturalist.org, accessed on 20 May 2025).

**Figure 2 tropicalmed-10-00225-f002:**
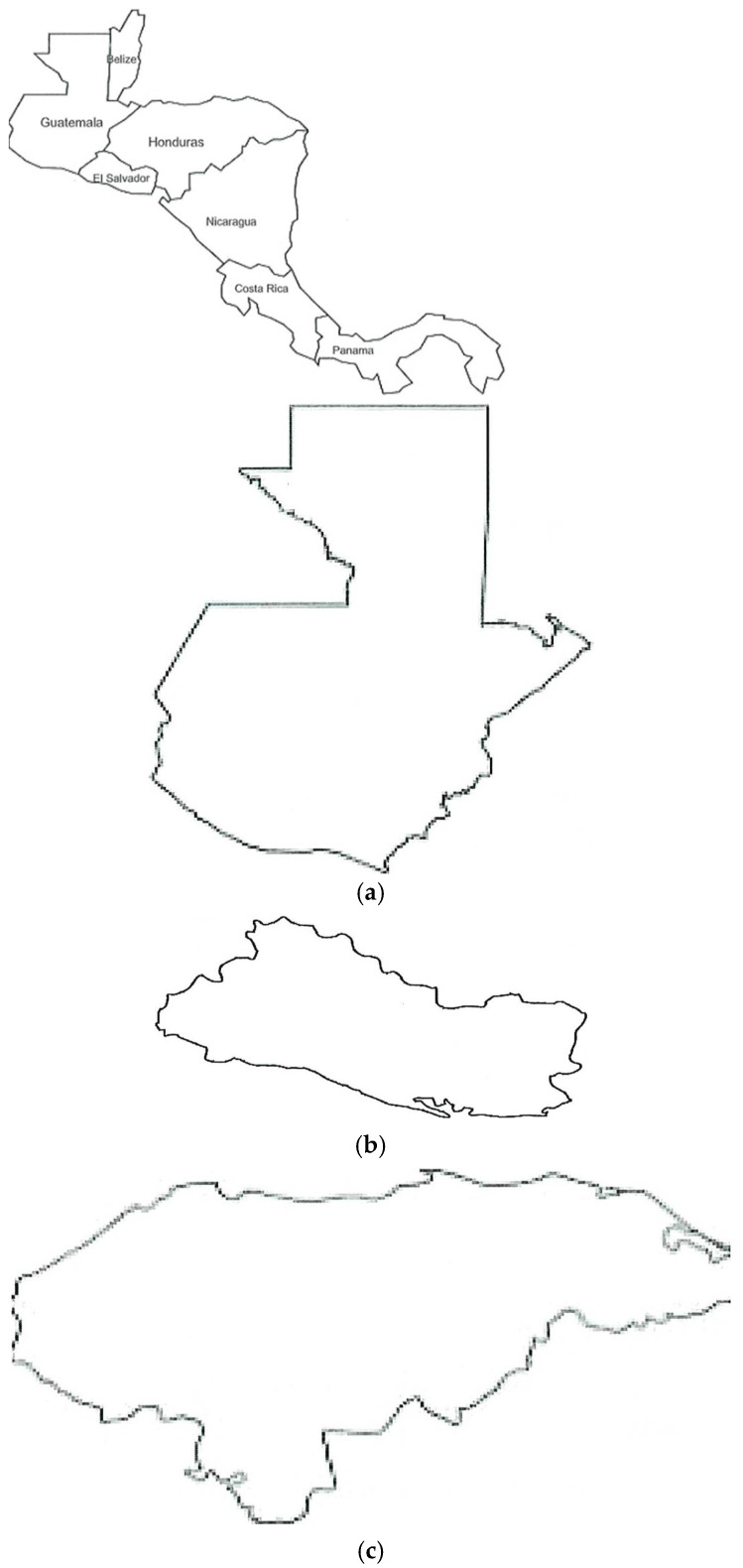
Relevant information on snakebites according to the reviewed studies in Central American countries. Adapted from GIS Geography (http://gisgeography.com/central-america-blank-map/, accessed on 20 May 2025) (**a**) GUATEMALA: Snakes reported as cause of snakebites: *B*. *asper*, *C. simus*, *M. nigrocintus*, *A. bilineatus*, *A. mexicanus*, *Porthidium ophiomegas*, and *C. durissus*. The main snake reported by ecological region: northern Guatemala (humid and close to the Caribbean Sea)*: B*. *asper*; central and southern Guatemala (dry areas and close to the Pacific coast): *C. simus*. Studies included in this review: Wellmann (2020) [28], Letona (2012) [31], Guerra-Centeno (2016) [32], Yee-Seuret (2012) [29], and Maltez (1994) [30]. The most commonly reported symptoms were hemorrhagic and coagulopathic. In severe cases, the brain, liver, and kidney were affected. Populations most affected: adolescents (10–19 years old) and young adults, mostly men dedicated to farming or living in rural areas. Body areas bitten most frequently: lower limbs. Mean time to reach a health facility after snakebite: 5.6 h (Guerra-Centeno, 2016 [32]). (**b**) EL SALVADOR: Located in the dry Pacific corridor of Central America. Main cause of snakebite envenoming: *Crotalus simus*. Other species with epidemiological importance: *Porthidium ophriomegas*, *Cerrophidiun wilsoni*, and *Micrurus nigrocinctus*. Mean snakebites per year (2014–2019): 188.3 cases [42]. Official records registered 4 deaths (for the period 2014–2019) [41,43]. Median age for the victims was 28 for men and 27 for women (older than those in the rest of Central America). Most cases came from rural locations and were men. (**c**) HONDURAS: Snakes reported as cause of snakebites: *B. asper*, *C. simus*, *Bothrops nasutus*, *Bothrops marchi*, *Micrurus* sp., and non-identified species of *Crotalus* sp. The main snake reported by ecological region: along the Caribbean Coast: *B asper*; central and southern Honduras: *C. simus* and *Micrurus* sp. (less than 5% of cases). Studies included in this review: Matute-Martínez (2016) [33], Laínez Mejía (2017) [34], Izaguirre Gonzalez (2017) [35], Ponce Orellana (2016) [36], Pinto L.J. (2019) [37], Cerrato Contreras (1989) [38], Cerna Rodríguez (2023) [39], and Aleman 2011 [40]. Most reported symptoms were hemorrhagic manifestations, tachycardia, dyspnea, pain, edema, and neurological symptomatology. One study reported pleural effusion and hemothorax. Populations affected most: mean of 18 years (young adults), men dedicated to farming or living in rural areas (students). Body areas bitten most frequently: lower limbs. Mean time to reach a health facility after the snakebite: 3–5.4 h (Matute-Martínez, 2016 [33]). (**d**) NICARAGUA: Main venomous species: *Crotalus simus* and *Bothrops asper* [54]. Snakebites were reported with most frequency in December, with the lowest incidence in May. Most snakebite reports came from the east and center regions of the country. Snakebite incidence in 5 years was 56 per 100,000 inhabitants, with 34 deaths and a case fatality of 1% in the period 2005–2009 [51]. Reported symptoms: pain, edema, bleeding, paresthesia, and vomiting. The most affected group was men living in rural areas aged from 12 to 20 years. Mean time to reach the hospital: 8 h, with a mean of 4 days of hospital stay [55]. (**e**) COSTA RICA: Coasts in the Caribbean Sea and Pacific Ocean. Main cause of snakebites: *Bothrops asper*, associated with lowlands. The most affected groups were young adults (20–29, 40–49, and 50–59), farmers, and male individuals [44]. Some snakebites were associated with recreational activities or activities in peridomiciliary areas. From 1993 to 2006, there were 48 fatalities due to snakebites. In 2012, the incidence rate was 9.44 and 10.76 per 100,000 inhabitants. The incidence of snakebites had two annual peaks: May–July and October–November. Snakebite management, raising public awareness, and the development of antivenom production were found to be national priorities. (**f**) PANAMA: Coasts in both oceans (narrowest area in the Americas). A publication in 1953 identified *Bothrops* sp. as responsible for venomous snakebites, recording 27 cases (7 with fatal outcomes [47]). Considered the country with the highest incidence of snakebites in Central America, with venoms causing hemorrhages, coagulopathic, edemas, myotoxic effects, and dermonecrosis. Severe cases led to cardiovascular shock and acute kidney injury [49,50]. Higher incidence during the rainy season. Victims reported damage to feet, toes, and hands [51], depending on the area affected by the bite. (**g**) BELIZE: Part of the hot, humid Caribbean coastline of Central America. No specific publications were found about Belize. According to local educational material, the most common venomous snakes include *Bothrops*, Crotalids, and Elapidae snakes. A recent snakebite in a 14-year-old boy affected his leg, causing edema, reddening, and local lesion.

**Table 1 tropicalmed-10-00225-t001:** Mortality of snakebite envenoming between 1990 and 2019 in Central America.

Country	Mortality 2019 Number of Individuals (Estimate)	Death Age Standardized Rate Per 100,000 in 2019 (DASR)	% of Change from 1990 to 2019 for DASR *	Years of Life Lost (YLLs) Age Standardized Rate Per 100,000 in 2019	% Change 1990–2019 (YLL) **
Guatemala	10 (7.9–13)	0.08 (0.06 to 0.10)	Increases [82%]	2.97 (0.22 to 3.71)	Increase [47%]
El Salvador	1.3 (<1–1.9)	0.02 (0.1 to 0.03)	Increases [222%]	0.88 (0.49 to 1.3)	Increase [150%]
Honduras	5.3 (3.0–8.4)	0.07 (0.04 to 0.12)	Decreases [53%]	2.68 (1.59 to 4.33)	Increase [65%]
Nicaragua	7.9 (4.9–10.0)	0.15 (0.10 to 0.19)	Decreases [51%]	5.95 (3.53 to 7.88)	Increase [57%]
Costa Rica	3.9 (2.9–52)	0.08 (0.06 to 0.10)	Decreases [20%]	2.4 (1.75 to 3.24)	Decrease [23%]
Panama	14 (11–19)	0.35 (0.26 to 0.46)	Decreases [50%]	14.96 (11.21 to 19.81)	Decrease [50%]
Belize	<1 (<1–1.1)	0.27(0.22 to 0.33)	Increases [444%]	10.38 (8.39 to 12.51)	Increase [372%]

Adapted from GBD 2019 Snakebite Envenomation Collaborators [7]. * Confidence intervals are omitted to simplify the table. ** Confidence intervals are omitted to simplify the table.

**Table 2 tropicalmed-10-00225-t002:** Antivenoms produced in Costa Rica and Mexico with specificity for Central America snake venoms (produced in Costa Rica and Mexico).

Name of the Antivenom	Use for This Type of Venoms	Presentation	Commercialized in the Region	For use in This Type of Medical Facilities	Country of Production
Coral ICP	Most coral snakebites of Central America	Liquid	Yes	Hospitals	Costa Rica
Polival ICP	Most viperine bites venoms	Liquid	Yes	Hospitals	Costa Rica
Lyophilized Polivalent	Viperine bites	Lyophilized formulation	Yes	Hospitals	Costa Rica
Antivipmyn	Pit-viper bites (crotalids)	Liquid and lyophilized formulations	Yes	Hospitals	Mexico
Birmex	Viperine bites	Lyophilized formulation	Yes	Hospital	Mexico

Sources: (https://icp.ucr.ac.cr/en/services-and-products/products-human-use, accessed on 20 May 2025); https://birmex.gob.mx/, accessed on 20 May 2025; https://www.miamidade.gov/fire/library/antivenom-species-covered.pdf, accessed on 20 May 2025.

**Table 3 tropicalmed-10-00225-t003:** Illustrative studies on snakebites in Central America. Adapted from studies conducted from 1950 to 2025.

Author	Country	Study Year(s)	# Cases	Mean Age (Years)	Common Symptoms	Time from Snakebite to Medical Facility	Main Snake Species Identified in the Study
Letona (2012) [31]	Guatemala	2002–2010	7377	15	Local pain, edema, bleeding	Not specified	*B asper*, *C. simus*, *A mexicanus*, *M. nigrocinctus*
Guerra-Otero (2016) [32]	Guatemala	2008–2013	305	25.2	Local pain, hemorrhages	5–6 h	*B. asper*, *C. simus*, *A. bilineatus*, *Micrurus* sp.
Yee-Seuret (2012) [29]	Guatemala	2008–2011	87	19	Local pain, edema	Not specified	*B. asper*, *C simus*
Izaguirre Gonzalez (2014) [35]	Honduras	2014–2015	36	15	Local inflammation, Gastrointestinal (GI) disorders	Not specified time	*C. simus*, *Micrurus sp*, *Botriechis marchi*, *B. nasutus*, *B. asper*
Laínez Mejía (2017) [34]	Honduras	2013–2015	84	28	Local pain and edema	5.4 h	*B. asper*, *C. simus*
Ponce Orellana (2016) [36]	Honduras	2015–2016	33	15	Local symptoms, GI, and hematological disorders	Not specified	*B. asper*, *Micrurus* spp.
Gutiérrez J.M. (2020) [41]	El Salvador	2014–2019	1130	15	Not specified	Not specified	*C. simus*
Chirino Molina (2025) [43]	El Salvador	2011–2022	1472	28	Not specified	Not specified	*C. simus*, *Porthidium ophriomegas*, *Cerrophidium wilsoni*
Fernandez P. (2008) [44]	Costa Rica	1993–2006	48 defunction cases	No Info	Not specified	No information	*B. asper*
Sasa (2020) [45]	Costa Rica	2012–2013	475	29	Coagulation disorders, pain, edema and local necrosis	1–3 h	*B. asper*, *Micrurus* sp.
Jutzy (1953) [47]	Panama	1953	23	No Info	Shock and hemorrhages	Not specified **Report 7 deaths**	*Bothropa atrox*
Pecchio (2019) [51]	Panama	2007–2018	390	No info	Local symptoms and hemorrhages	No information	*Bothrops asper*
Hansson (2010) [54]	Nicaragua	2005–2009	56	No info	No information	34 deaths were reported	*B. asper*, *C. simus*
Moreno Avellano (2000) [55]	Nicaragua	1997–1999	72	28	Pain, edema, paresthesia, hemorrhages	**8 h**	*Crotalus durissus*

## Data Availability

Databases are available online without any restrictions.

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
