# Peer review of "Snakebites in the Central American Region: More Government Attention Required"

_tropicalmed, 2025, doi:10.3390/tropicalmed10080225_

Round 1
Reviewer 1 Report
Comments and Suggestions for Authors
Dear Authors,
I read your manuscript entitled “Snakebites in the Central American Region Needing More Attention from the Governments”. You have tried to review the status of snakebite in Central American countries and your manuscript contains some valuable information. However, the manuscript lacks coherence, integrity and rigor of scientific writing. Moreover, I found it difficult to follow the manuscript for a number of reasons, including overly long sentences, poor grammar, multiple typo errors, insufficient information, and lack of clarity in the methodology (please see my comments below). I recommend extensive revisions on this manuscript with careful grammar checking.
GENERAL COMMENTS:
ABSTRACT:
- Page 1, Line 16: Please modify this part of the sentence: "Symptoms of local damage in the bite are such as…" TO “Symptoms of local damage in the bite site are such as…”
- Page 1, Lines 16-18: Please divide this sentence into two separate sentences, like this: “Symptoms of local damage at the bite site are such as edema, skin and muscle necrosis. In addition, cardiovascular system was affected with symptoms like hypotension, bleeding and coagulation disorders.
INTRODUCTION:
- Page 1, Line 28: You do not have to mention the acronym of Neglected Tropical Diseases, since this acronym “NTD” has not been used in the rest of text.
- Page 1, Lines 37-38: This sentence is incorrect and is not supported by a reference. The global status of snakebites is much higher. According to global estimates, 5.4 million people worldwide are bitten by snakes annually, with 1.8 to 2.7 million cases of envenomings. Please consider these references:
https://www.who.int/news-room/fact-sheets/detail/snakebite-envenoming
https://www.nature.com/articles/nrdp201763
- Page 1, Lines 39-40: What do you mean by stating this: Areas in the Americas and Australia are also “affected” by snakes…!!!! Please revise this sentence.
- Page 2, Line 52: At the end of the sentence, it is better to name the Central American countries, like this: “… with access to the Caribbean and the Pacific coasts, and include Belize, Costa Rica, El Salvador, Guatemala, Honduras, Nicaragua and Panama.”
- Page 3, Line 86: Please modify “primary bite…” TO “primary bite site”
- Page 4, Lines 111-112: The sentence has an ambiguous tone. The readers may misunderstand your sentence that an antivenom is a mixture of venoms or contains a mixture of venoms!!!! However, multiple venoms are used to immunize host animals for production of polyvalent antivenoms. Please revise the sentence carefully.
- Page 4, Line 112: Please provide a list of antivenoms (preferably a TABLE) that are produced in Central America. Potential readers of the manuscript will need to know the antivenoms produced in Central America (their names, their manufacturers, their nature (mono or polyvalent), their neutralization coverage (which snake venoms are neutralized by them), and their geographical coverage (in which countries are they available).
- Page 4, Line 133-134: The last sentence of the “INTRODUCTION” section poorly describes the objective of the study. “Get a sense of how much !!!!!!!!!” Please revise and express more logical objectives. Moreover, the last sentence of the “INTRODUCTION” section and the first sentence of “METHODS” section are better to be combined.
METHODS:
- Page 4, METHODS section: The nature of this review is not clarified. Is it a narrative review or a systematic review? If the latter is true, authors should describe their search strategy explicitly: Which search engines and which keywords were used? On what basis were the retrieved articles included in the final analysis and on what basis were they excluded? Other than published articles in peer-reviewed journals, how was the validity of the documents retrieved from university magazines or academic reports determined? The flow diagram of literature search (PRISMA flow diagram) should be illustrated and described.
RESULTS:
- In most parts of the RESULTS section; there are overly long sentences, and the overall text lacks integrity with many sentences losing their relevance to each other. Please carefully revise.
- Page 6, Line 232: Logically, “time” cannot be considered as an inconvenience. Perhaps you meant the “distance” between the location of snakebite incident and the hospital.
- Page 8, Line 312: What do you mean by this statement: “Panama was considered the country with highest incidence of snakebites”. Do you mean in Central America or the whole world?
- The manuscript needs some photos of common snakes responsible for snakebite events in Central America.
- A figure depicting the regions with higher frequency of snakebites in Central America is highly recommended. This can be a geographical figure depicting the countries of Central America and the regions with a higher likelihood of snakebite are marked with a symbol (e.g., a star)
DISCUSSION & CONCLUSIONS:
- Page 10, Line 376: What do you mean by stating “spoil of the inadequate antivenom”?!!!!
- Page 12, Line 443: What do you mean by stating “Accessibility to venoms….”?!!!! You probably meant “antivenoms” here, didn’t you?
GRAMMATICAL, WORDING & TYPO ERRORS:
There are several grammatical and typo errors throughout the text, including but not limited to:
- Page 2, Lines 60-63: Please carefully check the grammar.
- Page 2, Lines 71-72: This sentence: “According to the recorded clinical information this reflect a fraction of the real cases due to under-recording of cases.” is incomprehensible. It lacks coherence and relevance to the previous sentence.
- Page 3, Lines 70-72: Two sentences here lack relevance to each other and the second sentence is incomprehensible.
- Page 3, Line 84: What do you mean by this phrase: Organic systems!!!
- Page 3, Line 85: “Tissular” is not a common word, please modify it to tissue.
- Page 3, Line 88, 89: First part of the sentence lacks a verb.
- Page 3, Line 90-94: Please carefully revise the two sentences.
- Page 4, Line 115: Poor grammar: e.g., "a more complete neutralizations"!!!!
- Page 4, Line 131, 132: Please clarify that "his" refers to who in this part of the sentence: “Bothrops asper giving his role in envenoming”
- Page 5, Lines 176-177: Poor grammar, please revise: “Several inconvenient occurred”
- Page 5, Lines 187-188: Please modify “187 patients were male” to “One hundred eighty-seven patients were men”. In general, it is recommended to avoid starting a sentence with a numeral. Instead, it is better to spell out the number or you may rephrase the sentence. In addition, “male” is an adjective and in the context of this sentence it is better to use “men”.
- Page 6, Line 208: Please check the typo error: “tat species”?!!!
- Page 6, Line 206-211: The sentence is too long and suffers from grammatical errors. Please revise completely and divide it into 2-3 sentences.
- Page 6, Line 228: All species names should be italicized throughout the text. Hence, please italicize Crotalus simus.
- Page 6, Line 234: What do you mean by the word “media” in this sentence? You probably meant median, didn’t you?
- Page 6, Line 235: All species names should be italicized throughout the text. Hence, please italicize Bothrops asper.
- Page 6, Line 239: Poor grammar, please revise: “The patient improved his condition after treatment with antivenom”!!!
- Page 6, Line 243-245: Poor grammar, please revise.
- Page 6, Line 250-252: Poor grammar, please revise.
- Page 7, Line 262-263: Poor grammar, please revise: Snakebites are a obligatory notifiable event and its reported regularly.!!!
- Page 7, Line 274-275: Poor grammar, please revise: Patients included ages from 1-98 years…
- Page 7, Line 279-282: Poor grammar, please revise.
- Page 7, Line 292: “Neglected” within this sentence is not needed to be capitalized.
- Page 7, Line 302: “this” should be capitalized.
- Page 8, Lines 341-342: Poor grammar, please revise the sentence, like this: Moreno Avellano described people aged between 12 to 20 years old, living in rural areas and working in agriculture as the most affected groups with a clear male predominance.
- Page 8, Line 343: Poor choice of words. Please replace “The event” with “Snakebite”.
- Page 8, Line 347: Paresthesia does not need an “apostrophe s”.
- Page 8, Line 350: “He” in this sentence refers to whom? You probably meant “we”, didn’t you?
- Page 10, Lines 356-364: Too long sentence written in poor grammar. Please revise carefully and divide the sentence to at least 3 sentences.
- Page 11, Lines 369-373: Too long sentence written in poor grammar. Please revise carefully and divide the sentence to at least 2 sentences.
- Page 12, Line 431: Please modify “iess” to “less”.
The manuscript suffers from poor English language and requires extensive revisions.
Author Response
GENERAL COMMENTS:
ABSTRACT:
- Page 1, Line 16: Please modify this part of the sentence: "Symptoms of local damage in the bite are such as…" TO “Symptoms of local damage in the bite site are such as…
- =It has been modified to: ". Symptoms of local damage in the bite sites are those such as edema, skin and muscle necrosis."
- Page 1, Lines 16-18: Please divide this sentence into two separate sentences, like this: “Symptoms of local damage at the bite site are such as edema, skin and muscle necrosis. In addition, cardiovascular system was affected with symptoms like hypotension, bleeding and coagulation disorders.
- =The sentence was divided like this: "Symptoms of local damage in the bite sites are those such as edema, skin and muscle necrosis. In addition, the cardiovascular system was affected with symptoms like hypotension, bleeding and coagulation disorders''
INTRODUCTION:
- Page 1, Line 28: You do not have to mention the acronym of Neglected Tropical Diseases, since this acronym “NTD” has not been used in the rest of text.
- ="NTD It's not mentioned any more"
- Page 1, Lines 37-38: This sentence is incorrect and is not supported by a reference. The global status of snakebites is much higher. According to global estimates, 5.4 million people worldwide are bitten by snakes annually, with 1.8 to 2.7 million cases of envenomings. Please consider these references:
https://www.who.int/news-room/fact-sheets/detail/snakebite-envenoming
https://www.nature.com/articles/nrdp201763
="The references were considered and the information corrected : "According to recent WHO figures there are around 5.4 million cases of snakebites per year worldwide and from those close to 2.7 million develop clinical manifestations of envenoming[3,4].
- Page 1, Lines 39-40: What do you mean by stating this: Areas in the Americas and Australia are also “affected” by snakes…!!!! Please revise this sentence.
- It was revised to: "
. The Americas and Australia also report snakebites and their response to this health problem varies broadly [2, 3, 4] which is reflected in the surveillance of this program, the management of cases and mortality.
- Page 2, Line 52: At the end of the sentence, it is better to name the Central American countries, like this: “… with access to the Caribbean and the Pacific coasts, and include Belize, Costa Rica, El Salvador, Guatemala, Honduras, Nicaragua and Panama.
- =It was added the name of the countries: "Central America is a strip of land between North America and South America, with access to the Caribbean and the Pacific coasts, and includes Belize, Costa Rica, El Salvador, Guatemala, Honduras, Nicaragua and Panama."
- Page 3, Line 86: Please modify “primary bite…” TO “primary bite site”
- =Thank you. It has been modified to: "Cytotoxic effects cause extensive tissue destruction in the area around the primary bite site, compromising cutaneous, subcutaneous and muscular tissue [13,14]
- Page 4, Lines 111-112: The sentence has an ambiguous tone. The readers may misunderstand your sentence that an antivenom is a mixture of venoms or contains a mixture of venoms!!!! However, multiple venoms are used to immunize host animals for production of polyvalent antivenoms. Please revise the sentence carefully.
- =Thanks, it has been modified to: "
The health systems in Central America use antivenoms produced in laboratories using a mixture of the venoms of the most common venomous snakes in the Region. The process of anti-venom production requires de-inoculation of increasing doses of venom in laboratory animals to produce antiserums. The antiserums are then processed and purified for the use in individuals suffering snakebites and to neutralize the venom.[23]
- Page 4, Line 112: Please provide a list of antivenoms (preferably a TABLE) that are produced in Central America. Potential readers of the manuscript will need to know the antivenoms produced in Central America (their names, their manufacturers, their nature (mono or polyvalent), their neutralization coverage (which snake venoms are neutralized by them), and their geographical coverage (in which countries are they available).
- =A table with basic information on the type of venoms produced in the region was added in the following paragraphs (lines 128-130)
-
- Page 4, Line 133-134: The last sentence of the “INTRODUCTION” section poorly describes the objective of the study. “Get a sense of how much !!!!!!!!!” Please revise and express more logical objectives. Moreover, the last sentence of the “INTRODUCTION” section and the first sentence of “METHODS” section are better to be combined.
=This sentence was modified and more stress was given to the first methods paragraph: “It is important to know about the findings gathered by the different studies done in this region and their coincidences and differences”
- Methods
“The question to be answered is what are the main epidemiological features of the Snakebites in the countries of Central America and the time between the snakebite and the initial health care by the institutional system? Through the review of the literature produced about this problem in Central America we also identified gaps in the knowledge about snakebites and the symptomatology caused by them”
METHODS:
- Page 4, METHODS section: The nature of this review is not clarified. Is it a narrative review or a systematic review? If the latter is true, authors should describe their search strategy explicitly: Which search engines and which keywords were used? On what basis were the retrieved articles included in the final analysis and on what basis were they excluded? Other than published articles in peer-reviewed journals, how was the validity of the documents retrieved from university magazines or academic reports determined? The flow diagram of literature search (PRISMA flow diagram) should be illustrated and described.
- =It’s a narrative review, the quality of information varies between studies, some are University thesis and some others are publications , but it’s worthy to gather all this disperse information:
- “A narrative review of recent publications on Epidemiology and clinical characteristics of snakebites published on Central America snakebites was conducted to identify common species of venomous snakes, the symptomatology reported in the cases and the demographics of the population affected by the event..
- Different databases were reviewed including PUBMED, Scielos and Lilacs. Studies published in English and Spanish were included. Given the limited number of publications on the Region we included publications since 1950 to the current years and publica-tions retrieved from University Journals in the Region where they were available”
RESULTS:
- In most parts of the RESULTS section; there are overly long sentences, and the overall text lacks integrity with many sentences losing their relevance to each other. Please carefully revise.
- =”We have rewritten some sections to improve the flow and make shorter sentences”
- Page 6, Line 232: Logically, “time” cannot be considered as an inconvenience. Perhaps you meant the “distance” between the location of snakebite incident and the hospital.
- =The sentence was rephrased to explain the relationship time with distance and accessibility, but we also consider that time is critical in the distribution and metabolism of the toxins leading to more damage if the delay of the treatment delay”: We wrote: “ Some of the inconveniences for the opportune attention in the hospital was the distance to the health clinic and the factors related to accessibility to health care. These factors affected the critical time to be treated for their envenoming, because the venom action depends on the time the toxin is in circulation and in the local tissues. Two studies marked a range of 3-5 hours and two more provided a mean of 5.4 hours to reach the health care place [22, 36].”
- Page 8, Line 312: What do you mean by this statement: “Panama was considered the country with highest incidence of snakebites”. Do you mean in Central America or the whole world?
- ==Panama was considered by a local author as the country with highest incidence in Central America: “Panama was considered the country with highest incidence of snakebites in the Cen-tral American region by Velez (2017)”(page 7, line 241-42)
- The manuscript needs some photos of common snakes responsible for snakebite events in Central America.
- == Some common snakes illustrations’ were added in page 15 as figure # 2
- A figure depicting the regions with higher frequency of snakebites in Central America is highly recommended. This can be a geographical figure depicting the countries of Central America and the regions with a higher likelihood of snakebite are marked with a symbol (e.g., a star)
- A map was included with the country names and labeled sites where studies were conducted and capitals of the countries (page 14) as figure # 1
DISCUSSION & CONCLUSIONS:
- Page 10, Line 376: What do you mean by stating “spoil of the inadequate antivenom”?!!!!
- = It was changed to mean the need to have antivenoms in the areas reporting cases and providing adequate storage to the product, and using it before expiration:: “Two areas that need to be considered as priorities are the adequate supply of anti-venom where they are required (according to the epidemiological data) and the frequency of accidents caused by the different species to avoid problems and the discard of unused antivenom for lack of use, inadequate storage or for not being the required kind in the local clinic”.(lines 424-28)/
- Page 12, Line 443: What do you mean by stating “Accessibility to venoms….”?!!!! You probably meant “antivenoms” here, didn’t you?.
- =It was modified to antivenoms (thank you for noticing the lapsus) in lines 441-444 (page 12): “Not all patients receive treatment with antivenom and in optimistic calculations the coverage reaches less than 80%, one of the reasons for this deficit in the treatment is lack of clarity of the protocols for the management of these patients or the total absence of them in the medical facilities [25]”
GRAMMATICAL, WORDING & TYPO ERRORS:
There are several grammatical and typo errors throughout the text, including but not limited to:
- Page 2, Lines 60-63: Please carefully check the grammar.
- Lines 65-68. It’s now: “ Crotalus simus is reported as one of the main venomous snakes in inland and dry forest areas, and Coral snakes of the genus Micrurus (family Elapidae) are more relevant as a cause of snakebites along the Pacific coast, and it is associated with 1% of the snakebites. [11,12]”
- Page 2, Lines 71-72: This sentence: “According to the recorded clinical information this reflect a fraction of the real cases due to under-recording of cases.” is incomprehensible. It lacks coherence and relevance to the previous sentence.
- It was modified to: “Cases are not reported totally because of the limited accessibility to the health services and weakness of the epidemiological systems,. The specifics on the type of snake populations associated with every case are lost because of this deficient surveillance report and personal reactions associated with the encounter with these reptiles. [14, 15]
- Page 3, Lines 70-72: Two sentences here lack relevance to each other and the second sentence is incomprehensible.
The second sentence was modified (see previous answer) and the first one with basic information remain de same to improve the understanding.
- Page 3, Line 84: What do you mean by this phrase: Organic systems!!!
- It was changed to organs and systems, it was meant organs grouped in systems (Digestive, excretory). It’s meaning is more clear now.(page 3, line 92 now)
- Page 3, Line 85: “Tissular” is not a common word, please modify it to tissue.
- It was changed to tissue as suggested (page3 , line 93)
- Page 3, Line 88, 89: First part of the sentence lacks a verb
- The sentence has a verb now (line 96-97) “ Muscle damage is a frequent outcome as a result of local regeneration deficiencies, muscle loss and fibrosis”: .
- Page 3, Line 90-94: Please carefully revise the two sentences.
- The paragraph was slightly modified: ].”The deficient regeneration of the muscle remains a matter of research; one of the hypotheses is that Snake venom metalloproteinases (SVMP) -induced basement membrane damage, in micro vessels, muscle fibers and nerves, is the main culprit for the poor regenerative outcome [15 16]. The cardiovascular system is the most affected.I know is still complex but remain the idea of damage and cause (line 97-101)
- Page 4, Line 115: Poor grammar: e.g., "a more complete neutralizations"!!!!
- Thank you it was corrected to “a more complete neutralization” in line 126
- Page 4, Line 131, 132: Please clarify that "his" refers to who in this part of the sentence: “Bothrops asper giving his role in envenoming”
- It was intended as its. It was corrected . Thank you
- Page 5, Lines 176-177: Poor grammar, please revise: “Several inconvenient occurred”
- Thank you. It was pluralized to “ Several Inconveniences”
- Page 5, Lines 187-188: Please modify “187 patients were male” to “One hundred eighty-seven patients were men”. In general, it is recommended to avoid starting a sentence with a numeral. Instead, it is better to spell out the number or you may rephrase the sentence. In addition, “male” is an adjective and in the context of this sentence it is better to use “men”.
- =Both changes were introduced in the paragraph: “One hundred and eighty-seven patients were men (61.3% of the total) and dedicated to agricultural activities, 32% of the victims were farming while they experienced the event”
- Page 6, Line 208: Please check the typo error: “tat species”?!!!
- It was deleted. Thank you
- Page 6, Line 206-211: The sentence is too long and suffers from grammatical errors. Please revise completely and divide it into 2-3 sentences.
- =It was divided in smaller sentences:” The Guerra Centeno study identified that both hospitals had differences in the main snake species reported as responsible for the event (snakebite).
- The northern hospital (San Benito Peten) identified 88.5% of the clinical cases related to Bothrops asper snakebites
- . In the Southern Hospital Bothrops asper was reported with less frequency (13.3%) while Crotalus simus was responsible for 28.8%of cases, Agkistrodon bilineatus was associated to 37.5% of the events and coral snakes (Micrurus sp) were reported in 13.3% cases [31]
- Page 6, Line 228: All species names should be italicized throughout the text. Hence, please italicize Crotalus simus.
- It was italicized in line 257
- Page 6, Line 234: What do you mean by the word “media” in this sentence? You probably meant median, didn’t you?
- We meant Mean. A lapsus. Thank you
- Page 6, Line 235: All species names should be italicized throughout the text. Hence, please italicize Bothrops asper.
- It was italicized (it’s now in line 267)
- Page 6, Line 239: Poor grammar, please revise: “The patient improved his condition after treatment with antivenom”!!!
- Now in line 271, it was written as: “The patient had improvement of his condition after treatment including the use of anti-venom (37].
- Page 6, Line 243-245: Poor grammar, please revise.
- Thank you, it was rephrased to:” In Santa Teresa, a hospital located in the central dry valley of Comayagua, the reports of snakebites identified the causal snakes as Crotalus simus, Micrurus sp, Bothrops nasutus and Bothriechis marchi and a smaller number of snakebites by Bothrops asper.[35]
- Page 6, Line 250-252: Poor grammar, please revise.
- = It was revised to” The study was done in the community in extra-hospital settings and the community informants mentioned three deaths during 2011 among farmers having livestock in their farms [40]
- Page 7, Line 262-263: Poor grammar, please revise: Snakebites are a obligatory notifiable event and its reported regularly.!!!
- =Line 292 now. It was revised to: “Snakebites is an obligatory notifiable event, and snakebites are reported regularly.[42]”
- Page 7, Line 274-275: Poor grammar, please revise: Patients included ages from 1-98 years
- =It was changed to “ The patients age was in a range of 1-98 years and the median was 28 for men and 27 for women. Patients were older than in other Central American countries [42]. .…
- Page 7, Line 279-282: Poor grammar, please revise.
- =Revised to: “Chirino-Molino study identified as the responsible species for the snakebites Crotalus simus and other species such as Porthidium ophriomegas, Cerrophidion wilsoni and Micrurus nigrocinctus specifically recorded in the clinical records[43]
- Page 7, Line 292: “Neglected” within this sentence is not needed to be capitalized.
= Thank you. It’s not capitalized now
- Page 7, Line 302: “this” should be capitalized.
- = Thank you, it was a typo. Capitalization was removed
- Page 8, Lines 341-342: Poor grammar, please revise the sentence, like this: Moreno Avellano described people aged between 12 to 20 years old, living in rural areas and working in agriculture as the most affected groups with a clear male predominance.
- =it was modified to: “Moreno Avellano (2000) described the people aged between 12 to 20 years old, living in rural areas and working in agriculture as the most affected groups with ,a clear male predominance”.
- Page 8, Line 343: Poor choice of words. Please replace “The event” with “Snakebite”.
- It was replaced by a different construction: “The lower limbs were affected in 71% of the cases and the upper limbs in the rest”;
- Page 8, Line 347: Paresthesia does not need an “apostrophe s”.
- Thank you. It was a typo and it was removed
- Page 8, Line 350: “He” in this sentence refers to whom? You probably meant “we”, didn’t you?
- Yes, It was replaced by we in the manuscript
- Page 10, Lines 356-364: Too long sentence written in poor grammar. Please revise carefully and divide the sentence to at least 3 sentences
- =It was modified to: . Snakebites constitutes an important health problem in the Central American Region, but they still need to be studied and described in a more complete way.
- As a neglected tropical disease, snakebites suffer a lack of interest by most governments in Central America. The cases occurred in distant towns far from decision centers, the possibility to pose a pressing demand for solutions is low, which contributes to the postponement. of effective prevention and control policies[13]
- Page 11, Lines 369-373: Too long sentence written in poor grammar. Please revise carefully and divide the sentence to at least 2 sentences.
- Modified to: “ It's important to have a population better informed about the types of endemic snakes and the expected clinical damage of the ophidian accident. The knowledge of the risk posed by snakebites seems to be unevenly distributed in the population of the different countries.
- Page 12, Line 431: Please modify “iess” to “less”.
- = It has been corrected in line 489 page 11
- Thank you for the time and effort invested!!

Reviewer 2 Report
Comments and Suggestions for Authors
This is a great review of snakebites in Central America. It is an important health topic that does not get enough publicity. Some specific comments:
Italics needs to be used throughout article when talking about species.
Page 3, line 88. This sentence does not make sense. I'm not sure what it is trying to say.
Line 95: italics for Crotalus simus
Line 101: italics???
Page 5 line 188: yeas should be years
I do not think table 2 is necessary as this is described well in the text.
Comments on the Quality of English LanguageEnglish is overall good. There are some sentences that could use rewording as mentioned above.
Author Response
Italics needs to be used throughout article when talking about species.
=Thanks for the comments, we have reviewed the text to italicize the species name
Page 3, line 88. This sentence does not make sense. I'm not sure what it is trying to say.
=the sentence was modified and broken down for better understanding (line 96 now)
Line 95: italics for Crotalus simus
=it was italicized. Thank you.(line 102)
Line 101: italics???
=Yes, it has italics now in line 110
Page 5 line 188: yeas should be years
=thank you, it was corrected.
I do not think table 2 is necessary as this is described well in the text.
Thank you for this comment. I'm keeping it with some modifications suggested by a reviewer.

Reviewer 3 Report
Comments and Suggestions for Authors
This review compiles data from studies on snakebites in Central American countries, providing information on epidemiological and clinical aspects, and discussing the weaknesses in how governments of these countries address the problem. The subject is relevant for readers in this field, and the authors have gathered important information. However, in its current form, readers are likely to be interested only in the Introduction and Table 2.
The text in Section 3 is difficult to read due to a lack of fluency and an full of numerical and technical data. The Discussion section should provide a clearer overview and perform a comparative analysis among the countries studied, as well as with other regions of the world.
Major Comments
Table 1: Needs revision with clear explanations of the parameters presented. Readers outside the field will struggle to understand terms used. For example: what is the difference between “% of Change from 1990 to 2019” and “% change 1990-2019”? Country name correction: "Beliza" should be corrected to "Belize".
Line 136: Are there one or two research questions? It seems the study addresses two: What are the main epidemiological features of snakebites in Central American countries? What is the time between the ophidian accident and the initial health care provided by the institutional system? Although the epidemiological data were discussed, the issue of time to initial care was only addressed in four studies from four different countries. This is a limitation, but still important data. However, it was not sufficiently emphasized, especially given that it was one of the central questions of the study.
Additionally, the main clinical symptoms described in the analyzed studies—although thoroughly discussed—were not explicitly stated as a research question. Given its importance and level of detail in the results, the authors should highlight this as one of the study's objectives.
The manuscript frequently contains excessively long sentences (e.g., lines 103–107, 206–211, 264–269). Sentences with too many specific terms and data points need to be shortened for clarity.
Even shorter sentences, such as those in lines 235–239, are overloaded with information and should be rewritten for better readability.
Overall readability: The text is difficult to follow due to poor flow. While it is indeed challenging to write fluidly when dealing with dense technical and numerical data, this manuscript requires more attention in this regard.
To improve comprehension and enhance the manuscript, the authors should also consider including a schematic figure for each country-specific subsection. For example: a cartoon-style map with key data visually summarized around or above the map would help readers keep the information.
Minor Comments
Genus and species names should appear in italics throughout the manuscript.
Line 28: The inclusion of snakebites as a neglected tropical disease is not as recent as suggested—please revise.
YLLs (Years of Life Lost): This abbreviation needs to be defined when first introduced.
Line 72: Revise the sentence: “corresponds to the lack of information of the population of the snake populations” — this is unclear.
References 13, 17, and 18: Are these really about Crotalus simus? Please check. I suggest including this reference instead: DOI 10.1021/pr9008749.
Line 102: The title should be formatted as a distinct subheading.
Line 148: Why did the authors specifically draw attention to Panama? Please clarify.
Lines 153–159: The text here is very confusing and difficult to understand. Please rewrite for clarity.
Line 178: Remove the letter “A.”.
Author Response
Table 1: Needs revision with clear explanations of the parameters presented. Readers outside the field will struggle to understand terms used. For example: what is the difference between “% of Change from 1990 to 2019” and “% change 1990-2019”? Country name correction: "Beliza" should be corrected to "Belize".
=Thank you for your comment, we have make some additions to the table to make it easier to understand, and in the text we explain some of the terms. We corrected the country name Belize. Please find it in the document
Line 136: Are there one or two research questions? It seems the study addresses two: What are the main epidemiological features of snakebites in Central American countries? What is the time between the ophidian accident and the initial health care provided by the institutional system? Although the epidemiological data were discussed, the issue of time to initial care was only addressed in four studies from four different countries. This is a limitation, but still important data. However, it was not sufficiently emphasized, especially given that it was one of the central questions of the study.
Additionally, the main clinical symptoms described in the analyzed studies—although thoroughly discussed—were not explicitly stated as a research question. Given its importance and level of detail in the results, the authors should highlight this as one of the study's objectives.
=Yes, the study addresses two questions even when the quality of information is limited, and as you mention the symptomatology of patients has been stressed as an objective of the study. Please find it in methods, page 5 line 189
The manuscript frequently contains excessively long sentences (e.g., lines 103–107, 206–211, 264–269). Sentences with too many specific terms and data points need to be shortened for clarity.
Even shorter sentences, such as those in lines 235–239, are overloaded with information and should be rewritten for better readability.
= Long sentences have been modified in the text to increase understanding; they have been shortened and clarified as possible. Please see the text. Thank you is a very constructive observation.
Overall readability: The text is difficult to follow due to poor flow. While it is indeed challenging to write fluidly when dealing with dense technical and numerical data, this manuscript requires more attention in this regard.
=Some paragraphs were modified to improve the flow and overall understanding.
To improve comprehension and enhance the manuscript, the authors should also consider including a schematic figure for each country-specific subsection. For example: a cartoon-style map with key data visually summarized around or above the map would help readers keep the information
=Thank you for the suggestion, we included a map of the region with some of the places where the studies were conducted, the name of the countries and their relation with the reference areas of the Caribbean sea and the Pacific ocean with the importance in the ecology of the snakes. We included ilustrations of the main snakes in the region.
Minor comments
Genus and species names should appear in italics throughout the manuscript.
=They have been italicized throghout the manuscript. Thanks.
Line 28: The inclusion of snakebites as a neglected tropical disease is not as recent as suggested—please revise.
= We identified the year 2017 as the year of inclusion and it was posted without any other qualification, with its reference.
YLLs (Years of Life Lost): This abbreviation needs to be defined when first introduced.
=It was posted previous to the table with its definition. Thank you
Line 72: Revise the sentence “corresponds to the lack of information of the population of the snake populations” — this is unclear.
=it was changed to "The specifics on the type of snake populations associated with every case are lost because of this deficient surveillance report and personal reactions associated with the encounter with these reptiles
References 13, 17, and 18: Are these really about Crotalus simus? Please check. I suggest including this reference instead: DOI 10.1021/pr9008749.
=Thank you again, we adopted your suggested reference. We deeply appreciated.
Line 102: The title should be formatted as a distinct subheading.
=It was formatted as a subheading too.Thanks
Line 148: Why did the authors specifically draw attention to Panama? Please clarify.
=We eliminated the difference but until early in the 1900's Panama was considered geographically as part of South America. We agreed to avoid any special attention
Lines 153–159: The text here is very confusing and difficult to understand. Please rewrite for clarity.
=It was rewritten to: "
The different authors provided very similar information when they presented the demographics of the victims of snakebites (similar patterns).A total of 32 papers were reviewed in this comprehensive narrative review.
There was no scientific confirmation of the species causing the envenoming but only self-report of the common names for the snake and frequently that information was also absent. The symptoms are described in a similar way across different country studies, and some identify patterns of ophidian accident and responsible species according to ecological and climactic areas."
Thank you for all your observations
Line 178: Remove the letter “A.”.

Reviewer 4 Report
Comments and Suggestions for Authors
The submitted manuscript is a review of literature related to snakebite in Central America. Based on the title and abstract, I thought that the authors would summarize the most important aspects of snakebite and follow up care based on the studies they have compiled. Based on the results, they will then recommend practices that governments in the countries studied should consider. Although the authors collated data from published scientific studies as well as university and local sources, the overall quality of the study is quite inconsistent.
The work contains a large number of typos (missing punctuation, capitalization, unfinished parentheses, Latin names without italics, etc.), and the reading of the text is very disturbed. Also, the division into paragraphs in some places makes no sense at all. The text feels unfinished, in some parts it looks more like a first draft of a student's work.
What is a major problem for me, I found a paper by one of the authors on a similar topic: Fernández C, E.A., Youssef, P. Snakebites in the Americas: a Neglected Problem in Public Health. Curr Trop Med Rep 11, 19-27 (2024). https://doi.org/10.1007/s40475-023-00309-5. This paper compiles similar data, though it is cited in this study but nowhere mentioned. I assume that this study follows up on and extends that work, and that it is not self-plagiarized. In that case, it would be most appropriate to mention it here and describe how the studies differ or relate to each other.
The introduction is rather inconsistent in style and content. In some places there is a longer paragraph with a single citation, in others a reference is made to a specific study (although this is not strictly necessary for the text) and at the end of the paragraph there are two citations. I don't understand the structure of the introduction; paragraphs jump from topic to topic. There is no indication of which countries specifically are the focus of the review, conversely the information on where Central America is located is completely unnecessary, etc... There is no general information on the species of snakes that live in this area, just for the toxicity significant groups would suffice. Table 1 would be useful if the reader understood what it was saying. First, it is called "Global mortality..." but the data is only for the study area. It is not clear from the description what the variables are. E.g. "Mortality 2019", is this the number of people per year or some conversion to population? The abbreviation YLLs is not explained (even in the text). In the fourth and sixth columns, it is not clear what the numbers mean, somewhere the percentages are before the brackets, somewhere in the brackets. The use of a minus sign for percentages is misleading, rather the trend should be expressed in words (decreasing, increasing, stable) or by a symbol. Clear aims should be stated at the end of the introduction; a question at the beginning of the methodology is inadequate.
The results section does not indicate how many papers were included in the study. Table 2 includes only some papers, as I understand it. It would be useful to add a complete table with all the papers used, e.g. as Supplementary materials. The caption of Table 2 says 1980-2020, but there are studies from 1953-2022 (2025). Would it be possible to add the species of snakes involved in the bites (if mentioned in the paper) to the table as well? The results are broken down by country and described in detail. They contain important information, but the text lacks a coherent structure. Again, there is a problem with the writing of Latin names. If you are giving a species name for the first time, the Latin name should be in full, only then can the abbreviation be used. E.g. on line 178 it is "A. Bilineatus bihaentis" (a typo for bilineatus), only on line 210 it is "Agkistrodon bilineatus" (here again without italics). On line 180, the species is "A. mexicanus", but the full name is missing (I assume it is Atropoides mexicanus). Line 281 lists the species "Parthenium ophriomegas", what is it? Is this a typo and should it have been correctly Porthidium ophyomegas (BOCOURT 1868)?
The discussion is basically non-existent. It is a general summary, there are only four citations, no specific recommendations for the governments of the countries under review. The main message of this study is not clear from this.
The topic of your manuscript is very interesting and is still a current issue. Therefore, I regret that the processing and presentation of your results is not of sufficient quality for publication.
Comments on the Quality of English LanguageThe quality of the English is basically not bad. In some places similar words are repeated which could be replaced by equivalent expressions. What I don't understand is the capitalization. In chapter headings I could understand it, that is if the style was uniform. And I have a problem with writing the scientific names of snakes or families. For example, "Viperidae species" is not quite right. Better to write: Viperids species, Family Viperidae, Species of family Viperidae, etc. Furthermore, use the term Ophidians in the abstract. I know that "ophidian accidents" is often used in Central and South America. However, as a designation of a taxonomic group, this is misleading. In Latin, Ophidia refers to the group that includes modern snakes and extinct lineages that are probably most closely related to snakes. The term "snakes" is sufficient; if you want to use a more scientific name, "Serpentes" is more appropriate.
Author Response
The submitted manuscript is a review of literature related to snakebite in Central America. Based on the title and abstract, I thought that the authors would summarize the most important aspects of snakebite and follow up care based on the studies they have compiled. Based on the results, they will then recommend practices that governments in the countries studied should consider. Although the authors collated data from published scientific studies as well as university and local sources, the overall quality of the study is quite inconsistent.
= Thank you for this first comment. This manuscript as you mentioned tried to identify what has been written in Central America about snakebites. There is much more to be written. In the discussion and conclusions we include some of our issues on how to improve the management of the problem overall, it's a work in process I think and there is much to do. With your comments and the ones of the other reviewers we're trying to get more quality to our work.
The work contains a large number of typos (missing punctuation, capitalization, unfinished parentheses, Latin names without italics, etc.), and the reading of the text is very disturbed. Also, the division into paragraphs in some places makes no sense at all. The text feels unfinished, in some parts it looks more like a first draft of a student's work.
=We have tried to remove many of the typos you mentioned and make the reading easier and with a better flow, and working with paragraph contents. We expect to have improved the text.
What is a major problem for me, I found a paper by one of the authors on a similar topic: Fernández C, E.A., Youssef, P. Snakebites in the Americas: a Neglected Problem in Public Health. Curr Trop Med Rep 11, 19-27 (2024). https://doi.org/10.1007/s40475-023-00309-5. This paper compiles similar data, though it is cited in this study but nowhere mentioned. I assume that this study follows up on and extends that work, and that it is not self-plagiarized. In that case, it would be most appropriate to mention it here and describe how the studies differ or relate to each other.
=Important observation, Snakebites in the Americas a Neglected Problem in Public Health explores the problem in the Americas region with special interest in the presence of epidemiological surveillance in the different countries. In the current manuscript we focused in the Central American countries more in depth and with particular interest in the clinical manifestations of envenoming in the patients and the similarities/differences in the local studies and their relationship with the ecological characteristics of the venomous snakes in the region. Differences in depth and focus can be found in both documents. We have included a paragraph mentioning the previous work of the authors with the topic and the Central American Region (lines 81-83): "A previous study from one of the authors found the difficulties in surveillance and how local publications are sometime the only source of information. We expand our search in those additional publications [14]" as acknowledgement of that fact.
The introduction is rather inconsistent in style and content. In some places there is a longer paragraph with a single citation, in others a reference is made to a specific study (although this is not strictly necessary for the text) and at the end of the paragraph there are two citations. I don't understand the structure of the introduction; paragraphs jump from topic to topic. There is no indication of which countries specifically are the focus of the review, conversely the information on where Central America is located is completely unnecessary, etc... There is no general information on the species of snakes that live in this area, just for the toxicity significant groups would suffice.
=During the first review we have corrected many of the problems you mentioned. Paragraphs are shorter and flow better. Countries in the region are mentioned before we describe the studies done in every one. The snake families are described in the introduction briefly and additions are done throughout the text.
Table 1 would be useful if the reader understood what it was saying. First, it is called "Global mortality..." but the data is only for the study area. It is not clear from the description what the variables are. E.g. "Mortality 2019", is this the number of people per year or some conversion to population? The abbreviation YLLs is not explained (even in the text). In the fourth and sixth columns, it is not clear what the numbers mean, somewhere the percentages are before the brackets, somewhere in the brackets. The use of a minus sign for percentages is misleading, rather the trend should be expressed in words (decreasing, increasing, stable) or by a symbol. Clear aims should be stated at the end of the introduction; a question at the beginning of the methodology is inadequate.
=Table 1 had some modifications: the title was modified (no global), the columns explain better the contents. The term YLL was explained in the paragraphs preceding the table. The words reduce or increased were included for the fourth and six column to make self-explanatory.
About the introduction final paragraph expressed the very general idea for the study, and the research question was mentioned in the first paragraph of Methodology (in more detail).
The results section does not indicate how many papers were included in the study. Table 2 includes only some papers, as I understand it. It would be useful to add a complete table with all the papers used, e.g. as Supplementary materials. The caption of Table 2 says 1980-2020, but there are studies from 1953-2022 (2025). Would it be possible to add the species of snakes involved in the bites (if mentioned in the paper) to the table as well? The results are broken down by country and described in detail. They contain important information, but the text lacks a coherent structure.
=Thank you for this comment. We included the number of studies (32) in this review (line180 page 6). The table includes studies from the different countries and regions within the countries. Some studies had a different method and they will grouped for a next publication (later). We included in the table those with similar information and identifying (most of them) demographics and type of snake. The information of snakes was included in the table (table # 3 now) lines 381-383 in page 10.
The caption was corrected according to the chronology of the studies (since 1950).
The results are broken down by country and described in detail. They contain important information, but the text lacks a coherent structure. Again, there is a problem with the writing of Latin names. If you are giving a species name for the first time, the Latin name should be in full, only then can the abbreviation be used. E.g. on line 178 it is "A. Bilineatus bihaentis" (a typo for bilineatus), only on line 210 it is "Agkistrodon bilineatus" (here again without italics). On line 180, the species is "A. mexicanus", but the full name is missing (I assume it is Atropoides mexicanus). Line 281 lists the species "Parthenium ophriomegas", what is it? Is this a typo and should it have been correctly Porthidium ophyomegas (BOCOURT 1868)?
= We hope the coherence is much improved after this review. Latin names for scientific notation were corrected throughout the text. We avoided to use abbreviations the first time we introduced a scientic name, and corrected the mispelled names (page 6, lines 205-207, and page 8, line 210).
The discussion is basically non-existent. It is a general summary, there are only four citations, no specific recommendations for the governments of the countries under review. The main message of this study is not clear from this.
The discussion was expanded with some new ideas and similar experiences elsewhere, and more citations were added. The context of snakebites as neglected disease was stressed and the problems into influencing public policies and the government immediate actions.
The topic of your manuscript is very interesting and is still a current issue. Therefore, I regret that the processing and presentation of your results is not of sufficient quality for publication.
Thank you for your interest in our topic. With the editing and addition to the text we hope to improve the quality of the manuscripts. Thanks again.

Round 2
Reviewer 3 Report
Comments and Suggestions for Authors
The authors have improved the text by modifying some of the previously highlighted sentences. As noted above, the English is generally understandable; however, there are still issues with punctuation and some sentences would benefit from improved grammar and phrasing.
The added figure (Figure 1), however, does not address the objective of my previous suggestion: “authors should also consider including a schematic figure for each country-specific subsection. For example: a cartoon-style map with key data visually summarized around or above the map would help readers keep the information." The "key data" refers to the information gathered and discussed in the text regarding snakebite accidents, not the regional divisions of each country, as depicted in the figure presented.
In light of these points, I recommend another round of revision, now as minor revisions.
Author Response
July 27, 2025.
Dear Reviewer, 3:
Thanks for your time and effort invested in this new review. In the following paragraph we address your current comments:
The authors have improved the text by modifying some of the previously highlighted sentences. As n to oted above, the English is generally understandable; however, there are still issues with punctuation and some sentences would benefit from improved grammar and phrasing.
=Thank you for the comment. We review the text again to improve phrases and grammar.
The added figure (Figure 1), however, does not address the objective of my previous suggestion: “authors should also consider including a schematic figure for each country-specific subsection. For example: a cartoon-style map with key data visually summarized around or above the map would help readers keep the information." The "key data" refers to the information gathered and discussed in the text regarding snakebite accidents, not the regional divisions of each country, as depicted in the figure presented.
=We prepare the Figure 1 presenting each country with the most relevant information discussed in the paper, while keeping a general map of the area and simple physical drawing of each country to provide a geographical context. We hope this can give a quick sense of the information for each country. It’s in page 16, from line 562 to 662
Please find the complete document in the attachment.
Thanks again.
Dr. Eduardo Fernandez
Assistant Professor

Reviewer 4 Report
Comments and Suggestions for Authors
The authors have corrected the previous version of the manuscript, which covers an interesting and important topic but was not sufficiently well written. First of all, I would like to acknowledge the editing that the authors have done. They have completed the text of the manuscript, added a table and two figures, and substantially improved the discussion.
As far as the content and the structure of the subchapters are concerned, I have comments only on the introduction. This part could still use some editing. The first section on the issue of snakebites up to the table (line 49) is fine. At the end of this general section, it would be appropriate to say that the next part of the introduction already focuses on a specific area (Central America). The part of the text on lines 52-68 could already be a separate subchapter introducing the study site in terms of geography and the species that live there. The sentence on line 70 is unnecessary; it is not even clear whether it is a heading or what it is. Subchapter 1.2 is redundant, it is too short and the information is then listed for each group of snakes. Subsections 1.3, 1.4 and 1.5 are fine, only the heading for 1.4 is not in italics. At the end of the introduction, there should be a subsection/paragraph on the aims of the study, and there should also be information on the previous study that this work follows on from (this is now in subsection 1.1). In addition to this being the traditional placement of the aims (unless required as a separate chapter), this is recommended in the instructions to authors of this journal. The methodology chapter already describes how the study was specifically designed.
The authors have removed a large number of typos, but many still remain. I understand that the authors have tried to improve the content of the paper and respond to the reviewers' comments. However, I recommend re-reading the text thoroughly (for example, by an independent reader) and correcting all typos. I have a few comments on the formal aspect of the paper.
Table 1 has been corrected but is still not self-explanatory. Although abbreviations are explained in the text, this should also be in the table or its caption. In the third column the full name is given and then its abbreviation, then in the fourth column only this abbreviation is used. That is great and very appropriate. It would be appropriate to do the same for the other variable in the fifth column (YLLs) and it would be fine. In the second column, Mortality 2019, it is still not clear what the number means (number of people, some kind of coefficient?). The information that it is an estimate is not sufficient. I still have a problem with the values in the fourth and sixth columns (percent change). The trend and the percentage are fine, but the values in brackets don't make sense. If you have a value for 1990 and for 2019, you want to know by what percentage the 1990 value increased or decreased compared to 2019. So you have what percentage (absolute value without sign) and the trend (increase/decrease/stable). You can put specific values for the two years in brackets, percentages are meaningless in this case.
Table 2 should have the same heading format as Table 1. There should also be uniform naming of snake groups (here and in the text). Either use names for the subfamilies Viperinae and Crotalinae, or viperines/true vipers and pit vipers. Subfamilies and families should not be in italics.
For Table 3, consider listing all publications found. If there are 32 papers, the table should not be too large, and this would be appropriate for the presentation of the results. Otherwise, you would need to indicate on what basis you selected the examples for the table.
Figure 1 is not entirely appropriate, and the source is not given (or is it your own work?). A geographical map of the study area would be better, e.g. this one: https://www.britannica.com/place/Central-America.
Figure 2 has inconsistent image captions. Each picture should have the scientific name (and English name if you want), the family and the author of the original photo. It is not clear to me why the photos are not from the same source. For example, at inaturalist.org, where you get your coralsnake photo, all of these species are available and there are enough photos without copyright restrictions (various CC licenses). Then you would only need to list one source of photos in the caption.
You've certainly improved your writing of the Latin names of snakes. I don't insist on full names everywhere, use abbreviations for the most commonly mentioned ones, but be consistent in your text, as well as when writing families/subfamilies.
Otherwise, I like your work, after these minor problems are corrected, the work will be suitable for publication.
Author Response
St Catharines July 27, 2025
Dear Reviewer # 4,
Thank you for your comments, observations and suggestions. We highly appreciate the time and effort invested in reviewing our work.
About your current comments we have the following answers and actions.
The authors have corrected the previous version of the manuscript, which covers an interesting and important topic but was not sufficiently well written. First of all, I would like to acknowledge the editing that the authors have done. They have completed the text of the manuscript, added a table and two figures, and substantially improved the discussion.
= Thank you, we followed your advice
As far as the content and the structure of the subchapters are concerned, I have comments only on the introduction. This part could still use some editing. The first section on the issue of snakebites up to the table (line 49) is fine. At the end of this general section, it would be appropriate to say that the next part of the introduction already focuses on a specific area (Central America). The part of the text on lines 52-68 could already be a separate subchapter introducing the study site in terms of geography and the species that live there.
=Thanks. A line was included to introduce the specifics for snakebites in Central America (lines 60,61 in page 3). A section is open after it.
The sentence on line 70 is unnecessary; it is not even clear whether it is a heading or what it is.
= This sentence as heading was deleted
Subchapter 1.2 is redundant, it is too short and the information is then listed for each group of snakes.
=This subchapter was suppressed and the numbering was changed
Subsections 1.3, 1.4 and 1.5 are fine, only the heading for 1.4 is not in italics.
= It was italicized only the technical name
At the end of the introduction, there should be a subsection/paragraph on the aims of the study, and there should also be information on the previous study that this work follows on from (this is now in subsection 1.1). In addition to this being the traditional placement of the aims (unless required as a separate chapter), this is recommended in the instructions to authors of this journal. The methodology chapter already describes how the study was specifically designed.
= A paragraph was introduced about the previous study in lines.159-162 (page 5)
The aims of the study are presented too in lines.163-170 (page 5)
The authors have removed a large number of typos, but many still remain. I understand that the authors have tried to improve the content of the paper and respond to the reviewers' comments. However, I recommend re-reading the text thoroughly (for example, by an independent reader) and correcting all typos. I have a few comments on the formal aspect of the paper.
=An independent reader went over the text to reduce the risk of typos
Table 1 has been corrected but is still not self-explanatory. Although abbreviations are explained in the text, this should also be in the table or its caption.
In the third column the full name is given and then its abbreviation, then in the fourth column only this abbreviation is used. That is great and very appropriate. It would be appropriate to do the same for the other variable in the fifth column (YLLs) and it would be fine
=We explained the abbreviation in column 5
. In the second column, Mortality 2019, it is still not clear what the number means (number of people, some kind of coefficient?). The information that it is an estimate is not sufficient.
= It is the number of people, an explanation was included (# of individuals)
I still have a problem with the values in the fourth and sixth columns (percent change). The trend and the percentage are fine, but the values in brackets don't make sense. If you have a value for 1990 and for 2019, you want to know by what percentage the 1990 value increased or decreased compared to 2019. So you have what percentage (absolute value without sign) and the trend (increase/decrease/stable). You can put specific values for the two years in brackets; percentages are meaningless in this case.
=We eliminated the confidence intervals values in brackets to increase the clarity of the values. We explained the reason for elimination in the bottom of the table. Getting close to the absolute numbers is limited by the source we used (GBD)..
Table 2 should have the same heading format as Table 1. There should also be uniform naming of snake groups (here and in the text). Either use names for the subfamilies Viperinae and Crotalinae, or viperines/true vipers and pit vipers. Subfamilies and families should not be in italics.
=Heading format of table 2 is now similar to Table 1. Naming of snake bites is uniform and no italics were used.
For Table 3, consider listing all publications found. If there are 32 papers, the table should not be too large, and this would be appropriate for the presentation of the results. Otherwise, you would need to indicate on what basis you selected the examples for the table.
=An extended list of publications (with the 32 papers) was extended at the end of the paper as an Appendix # 1 (The original, shorter list was kept reducing the space in the main text)
Figure 1 is not entirely appropriate, and the source is not given (or is it your own work?). A geographical map of the study area would be better, e.g. this one: https://www.britannica.com/place/Central-America.
= The source is given now (https :// gisgeography.com/central-america-blank-map/, and the purpose was to provide a sketchy idea of the individual countries and provide key information presented in the paper text), there is information for every country
Figure 2 has inconsistent image captions. Each picture should have the scientific name (and English name if you want), the family and the author of the original photo. It is not clear to me why the photos are not from the same source. For example, at inaturalist.org, where you get your coralsnake photo, all of these species are available and there are enough photos without copyright restrictions (various CC licenses). Then you would only need to list one source of photos in the caption.
= We followed your advice and we gain more quality of images.: they all come from inaturalist.com.
You've certainly improved your writing of the Latin names of snakes. I don't insist on full names everywhere, use abbreviations for the most commonly mentioned ones, but be consistent in your text, as well as when writing families/subfamilies.
= Thank you for your guidance.
Otherwise, I like your work, after these minor problems are corrected, the work will be suitable for publication.
=Thanks again.
With our best regards
Eduardo Fernandez PhD
Corresponding author
